# A Comparative Assessment of Random Forest and k-Nearest Neighbor Classifiers for Gully Erosion Susceptibility Mapping

**Mohammadtaghi Avand [1], Saeid Janizadeh [1], Seyed Amir Naghibi [1], Hamid Reza Pourghasemi [2,*], Saeid Khosrobeigi Bozchaloei [3] and Thomas Blaschke [4]**

[1] Department of Watershed Management Engineering and Sciences, Faculty in Natural Resources and Marine Science, Tarbiat Modares University, Tehran 14115-111, Iran; mt.avand@modares.ac.ir (M.A.); janizadehsaeid@modares.ac.ir (S.J.); amir.naghibi@modares.ac.ir (S.A.N.)

[2] Department of Natural Resources and Environment Engineering, College of Agriculture, Shiraz University, Shiraz 71441-65186, Iran

[3] Department of Watershed Management, Faculty in Natural Resources, Tehran University, Tehran 14174-14418, Iran; s.khosrobeigi@ut.ac.ir

[4] Department of Geoinformatics – Z_GIS, University of Salzburg, 5020 Salzburg, Austria; Thomas.Blaschke@sbg.ac.at

\* Correspondence: hr.pourghasemi@shirazu.ac.ir; Tel.: +98-911-120-8810

**Abstract:** This research was conducted to determine which areas in the Robat Turk watershed in Iran are sensitive to gully erosion, and to define the relationship between gully erosion and geo-environmental factors by two data mining techniques, namely, Random Forest (RF) and k-Nearest Neighbors (KNN). First, 242 gully locations we determined in field surveys and mapped in ArcGIS software. Then, twelve gully-related conditioning factors were selected. Our results showed that, for both the RF and KNN models, altitude, distance to roads, and distance from the river had the highest influence upon gully erosion sensitivity. We assessed the gully erosion susceptibility maps using the Receiver Operating Characteristic (ROC) curve. Validation results showed that the RF and KNN models had Area Under the Curve (AUC) 87.4 and 80.9%, respectively. As a result, the RF method has better performance compared with the KNN method for mapping gully erosion susceptibility. Rainfall, altitude, and distance from a river were identified as the most important factors affecting gully erosion in this area. The methodology used in this research is transferable to other regions to determine which areas are prone to gully erosion and to explicitly delineate high-risk zones within these areas.

**Keywords:** gully erosion; random forest; KNN; geo-environmental factors; Robat Turk area

---

## 1. Introduction

Water erosion is considered as one of the basic causes of land destruction, particularly in watersheds [1]. Erosion and soil destruction are among the most significant consequences of water erosion. Soil erosion has been studied by various researchers; its consequences are very dangerous, and both the flourishing and the destruction of previous civilizations have been attributed to this phenomenon. Despite experts extensively studying gully erosion in the twentieth century, and previous work before the 1930s surveying the factors that influence its formation and expansion, gully erosion is still classified in a variety of ways in various regions of the world. Meanwhile, despite considerable research on soil erosion and its various dimensions, many aspects of this phenomenon remain unknown or are only vaguely understood [2].

Gully erosion is the devastating kind of erosion, which can become a hazard if human factors such as land-use change, grazing pressure, and inadequate farming are exacerbated. This type of erosion causes sediment production in the environment, and is one of the important signs of land destruction [3]. Gully erosion is an action in which surface soil is degraded and runoff concentrates within canals, thereby deepening the canals [4]. Gully erosion is a complicated and devastating process of water erosion, which begins with a depression in the land, leached by waterfalls, and develops by head cut [4–6]. Because of the damage caused by gully erosion, this phenomenon is a permanent threat to soil ecosystems, land, and economic stability [7–9]. In general, gully erosion has three main impacts: (1) eliminating valuable agricultural soil and reducing yield and soil fertility, (2) increased fluidity of surface water, and thus, a higher risk of flooding, and (3) sedimentation in reservoirs and dams, reducing their useful lifespan [10–12].

Under specific conditions, gully erosion can also cause landslides and debris flow [8,12]. Mechanisms such as penetration, piping, and underground flow can lead to soil dispersion and the collapse of piping walls, and eventually, gully erosion [1,13,14]. The development and growth of rill erosion can also lead to gully erosion [15–17]. Gully erosion is an important source of sediment production; 10 to 94 percent [18] of gully erosion occurs at the watershed scale [3]. Accordingly, understanding gully erosion is necessary to better manage erosion and to identify and prioritize areas which are susceptible to it in order to reduce the risks associated with it [3]. Various climatic, hydrological, and physical factors have been reported as the most important ones affecting gully erosion [3,19–21].

Our literature review revealed that different data mining and statistical methods of gully erosion assessment have been used, including classification and regression tree-CART [22], Logistic regression-LR [23–26], BF-Tree for gully headcut [27], and frequency ratio-FR [28]. Angileri et al. [29] used Stochastic Gradient Treeboost (SGT) to analyze and forecast the spatial occurrence of the rill-inter rill and gully erosion types in central-northern Sicily (Italy). They stated that SGT is a good theory to better clarify the relationships among erosion and environmental variables.

Accordingly, the purposes of the current research are (1) to model gully erosion susceptibility using two famous data mining techniques, namely RF and KNN, (2) to determine the importance of the geo-environmental/conditioning factors for gully erosion susceptibility mapping, and, finally, (3) to provide an applicable guideline for stakeholders to reduce gully damage in the study area. The main novelty of the current research is the application of the RF and KNN data driven methods for gully susceptibility mapping in order to compare both models for the first time. Therefore, the gully susceptibility map is an appropriate tool with which to understand the mechanism of gully erosion and to aid scientific planning and decision making. Also, this map is applicable to the management of land use, gully erosion risk, and sustainable development in the Robat Turk area.

## 2. Materials and Methods

### 2.1. Study Area

The Robat Turk Watershed is one of the sub-watersheds of the Shoor River, situated between Markazi and Isfahan Provinces, spanning from 33°42′ to 35°45′ N latitude and 50°46′ to 50°52′ E longitude (Figure 1). The area of Rabat Turk is 242 km$^2$, with a minimum altitude of 1807 m and a maximum altitude of 2723 m. The climate of the study area is arid and semi-arid; the annual rainfall is 213 mm [30]. Approximately 80% of the annual rainfall in this area occurs in December and April. Peak stream flows occur from February to June [30]. Land-use types in the study region include agriculture, bare land, and pastureland uses, whereby bare lands cover a large share of the region. Most of the gullies in the region are concentrated in the northern regions, where bare land and agricultural land are prevalent. Investigation of the morphometric properties of the gullies in two land uses (agriculture and rangeland) showed that they were relatively active for both. The shape of the gullies is concave and vertical, and there are soil fragments within the gully canal, as well as wall

cracks that cause the longitudinal profile to be convex in some cases. Also, the transverse profiles of each gully differ greatly, indicating that the gullies are active. The shape of the transverse profiles of the gullies varies widely, indicating that the gully is active as the walls are gradually destroyed and fall inwards [31]. Figure 2 depicts two instances of gully erosion in this area.

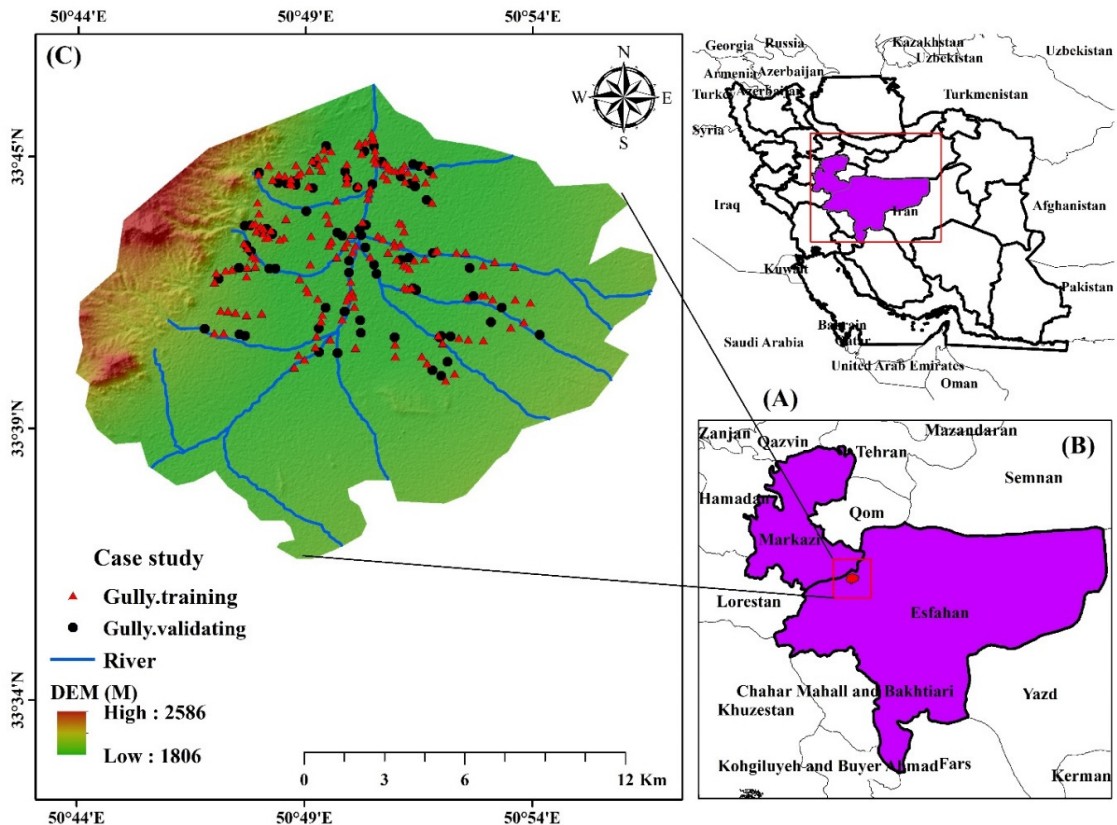

**Figure 1.** (**A**) the Markazi and Isfahan Provinces in Iran, (**B**) the study area in the Markazi and Isfahan Provinces, and (**C**) gully erosion situations with a DEM map of the Robat Turk Watershed.

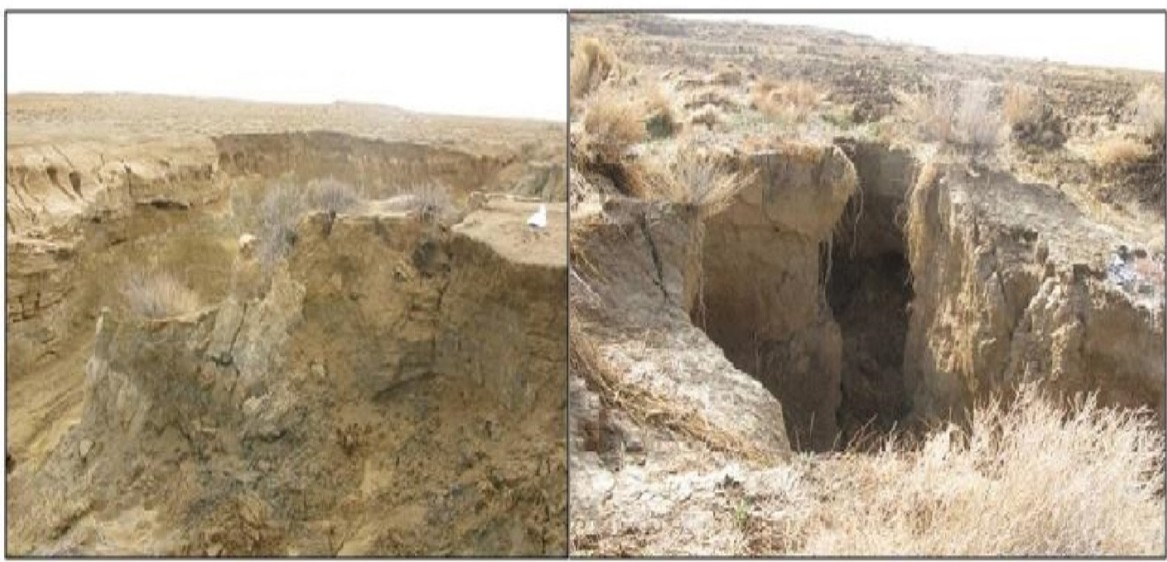

**Figure 2.** A picture of two gullies created in agriculture land (**right**) and range land (**left**) study area.

## 2.2. Methodology

### 2.2.1. Gully Dataset

To provide a gully erosion inventory map in the Robat Turk watershed, we first carried out field surveys to identify V- and U-shaped gully occurrences. This revealed a total of 242 gully locations in the study area [32]. These gullies mostly occur in plains and low slopes, where drainage density is high. In these areas, gully erosion and piping have developed due to the low vegetation and the high evaporation causing the formation of salt and gypsum in the soil [1,22]. To separate the gullies for training and validation purposes, a random partition algorithm was used [32–36]. Of the 242 gully locations and the same number of non-gully locations, 70% were used for the training stage and 30% for the validation stage [27]. Furthermore, the same number of non-gully erosion points were prepared for training and validation process [1,29,37,38]. Figure 3 displays a flow diagram of the methodology implemented in this study to create the gully erosion susceptibility maps.

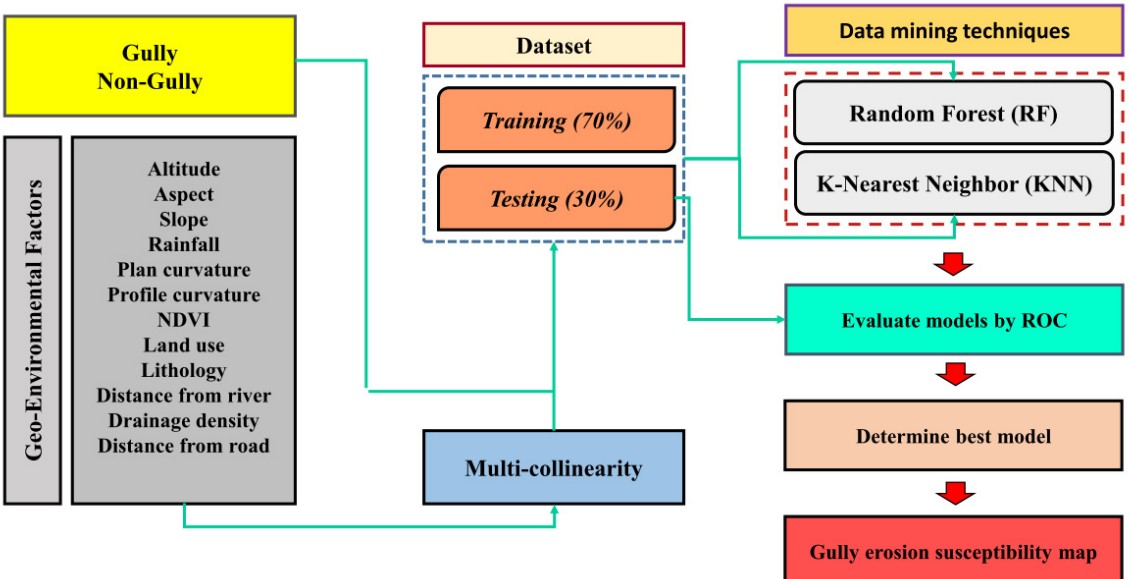

**Figure 3.** Flowchart of the research methodology used to provide the gully erosion susceptibility map.

### 2.2.2. Gully Erosion Geo-Environmental Factors

Based on previous studies [24,25,39–41] and the available data, hydrologic-geological-physiographic factors, including altitude, slope degree, slope aspect, plan curvature, profile curvature, distance to rivers, drainage density, distance to roads, lithology, land use, Normalized Difference Vegetation Index (NDVI), and annual mean rainfall were identified as important factors for gully erosion appraisals.

The Digital Elevation Model (DEM) of the study area was derived from ALOS PALSAR Global Radar Imagery with a spatial resolution of 12.5 m × 12.5 m [42] (Figure 4a).

Several layers, namely, slope degree, slope aspect, plan curvature, and profile curvature, were created using the ALOS PALSAR DEM. Slope is an influential variable in the gully erosion process due to its impact on surface flow and drainage density, and because it causes the expansion of the gully [41]. The ArcGIS 10.5 software (developed by Environmental Systems Research Institute (ESRI) located in Redlands, California, USA) was used to prepare the slope map (Figure 4b), which was subsequently classified into five classes 0–5, 5–12, 12–20, 20–30, and >30 degrees, according to Zabihi et al. [43].

Slope aspect is a significant variable due to its effect on the type of vegetation present in a given area. It controls the duration of sunlight, moisture, evaporation, and transpiration, and the distribution

of vegetation that indirectly affects the erosion process [44]. The aspect map was extracted from the DEM and classified into nine classes (Figure 4c) according to categorical features.

Useful geomorphological information and morphological land descriptions can be defined by analyzing the slope shape [40]. The plan and profile curvatures affect the convergence or divergence of the flow [45]. Plan and profile curvatures were prepared from the DEM with pixel size 12.5 m × 12.5 m [46] using ArcGIS 10.5, and were subsequently divided into three classes, namely, concave, flat, and convex (Figure 4d–e).

Gullies are always associated with drainage networks [1]. In order to survey the effect of the stream network on gully erosion, the distance to rivers and drainage density factors were used (Figure 4f–g). Surface runoff is high in areas in which the drainage density is high. Drainage density can also affect the drainage pattern in an area, and the development drainage density depends on many variables, such as the structure and nature of geological formation, soil features, vegetation conditions, penetration rate, and slope degree [47,48]. A drainage density map was developed in ArcGIS 10.5 using Line Density Tools. The distance to rivers factor was also determined using the Euclidean Distance Tools in ArcGIS 10.5 software.

Gully erosion depends on the lithological features of the surface material and the shape of the surface in Earth [25,49]. A geological map with a scale of 1:100,000 was used to prepare the lithology map [50]. The lithology map was divided into eight classes using ArcGIS 10.5 (Table 1, Figure 4h).

**Table 1.** Lithology available in Robat Turk watershed.

| Row | Code | Lithology | Geological Age |
|---|---|---|---|
| 1 | Qft2 | Low level pediment fan and valley terrace deposits | Quaternary |
| 2 | Plc | Polymictic conglomerate and sandstone | Pliocene |
| 3 | pCk | Dull green grey salty shales with subordinate intercalation of quartzitic sandstone (KAHAR FM; Morad series and Kalmard Formation) | Pre-Cambrian |
| 4 | Ekgy | Gypsum | Late Eocene |
| 5 | E2l | Nummulitic limestone | Eocene |
| 6 | pCmt2 | Low - grade, regional metamorphic rocks (Green Schist Facies) | Pre-Cambrian |
| 7 | OMql | Massive to thick - bedded reefal limestone | Oligocene-Miocene |
| 8 | Pd | Red sandstone and shale with subordinate sandy limestone (Dorud Formation) | Permian |

Land-use management has a significant impact on the geomorphology of slope stability and the incidence of gully erosion. Generally, vegetation-free areas and scattered areas have a higher sensitivity to erosion than those with good vegetation coverage [37,40]. The land use map for this study was generated using Landsat 8 (OLI) imagery [51] processed in the ENVI 5.4 software (developed by Harris Geospatial Solutions located in Broomfield, Colorado, United States). The land-use classes identified in the region are bare land, range land, and agriculture areas (Figure 4i).

The normalized difference vegetation index (NDVI) quantifies vegetation by measuring the difference between near-infrared (which vegetation strongly reflects) and red light (which vegetation absorbs). Areas covered in dense vegetation often have fewer instances of gully erosion [52]. The NDVI map used for this study was generated from Landsat 8 imagery collected on 15 June 2017 (Figure 4j). The NDVI value is computed by the following equation:

$$NDVI = (NIR - Red)/(NIR + Red) \qquad (1)$$

where near-infrared (NIR) is band 5 of Landsat 8 imagery within a wavelength range of 0.845–0.885 μm, and Red is band 4 of Landsat 8 imagery within a wavelength range of 0.63–0.68 μm. The range obtained from this index varies from −1 to +1, with positive numerical values for dense vegetation, zero and near numerical values for water areas and negative numerical values indicating low vegetation areas.

Some linear and manmade phenomena, such as roads and canals, can be susceptible to gully erosion [1,18,53]. Improper road construction disrupts natural drainage and, as a result, expands erosion. Consequently, in areas with bare soil, inadequate construction can exacerbate gully erosion [54]. Therefore, a distance-from-roads map was created using the Euclidean Distance tools in ArcGIS 10.5 software (Figure 4k), to be used in the creation of the gully erosion susceptibility map. In order to prepare rainfall data, three rain gauge stations (inside and outside the basin) were used. After checking the accuracy of different interpolation methods, the annual rainfall map of the Robat Turk watershed was provided by the Inverse Distance Weighting (IDW) method (Figure 4l). IDW is one of the most applicable and deterministic techniques of interpolation in the environmental sciences. IDW estimates are based on known locations nearby. The weights assigned to the interpolation points are the distance from the interpolation point. As a result, short distances are made to have more weight (therefore, more impact) than distant points. Well known sample points indicate that they are controlled by each other [55].

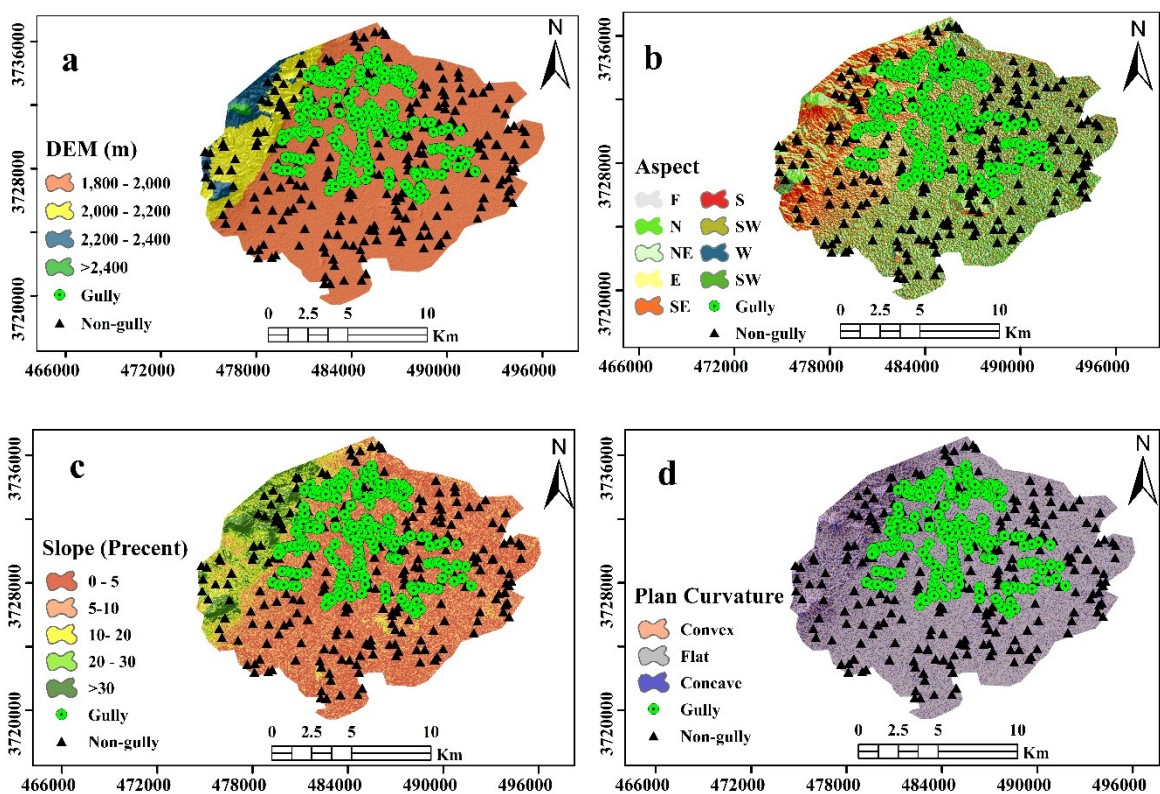

**Figure 4.** *Cont.*

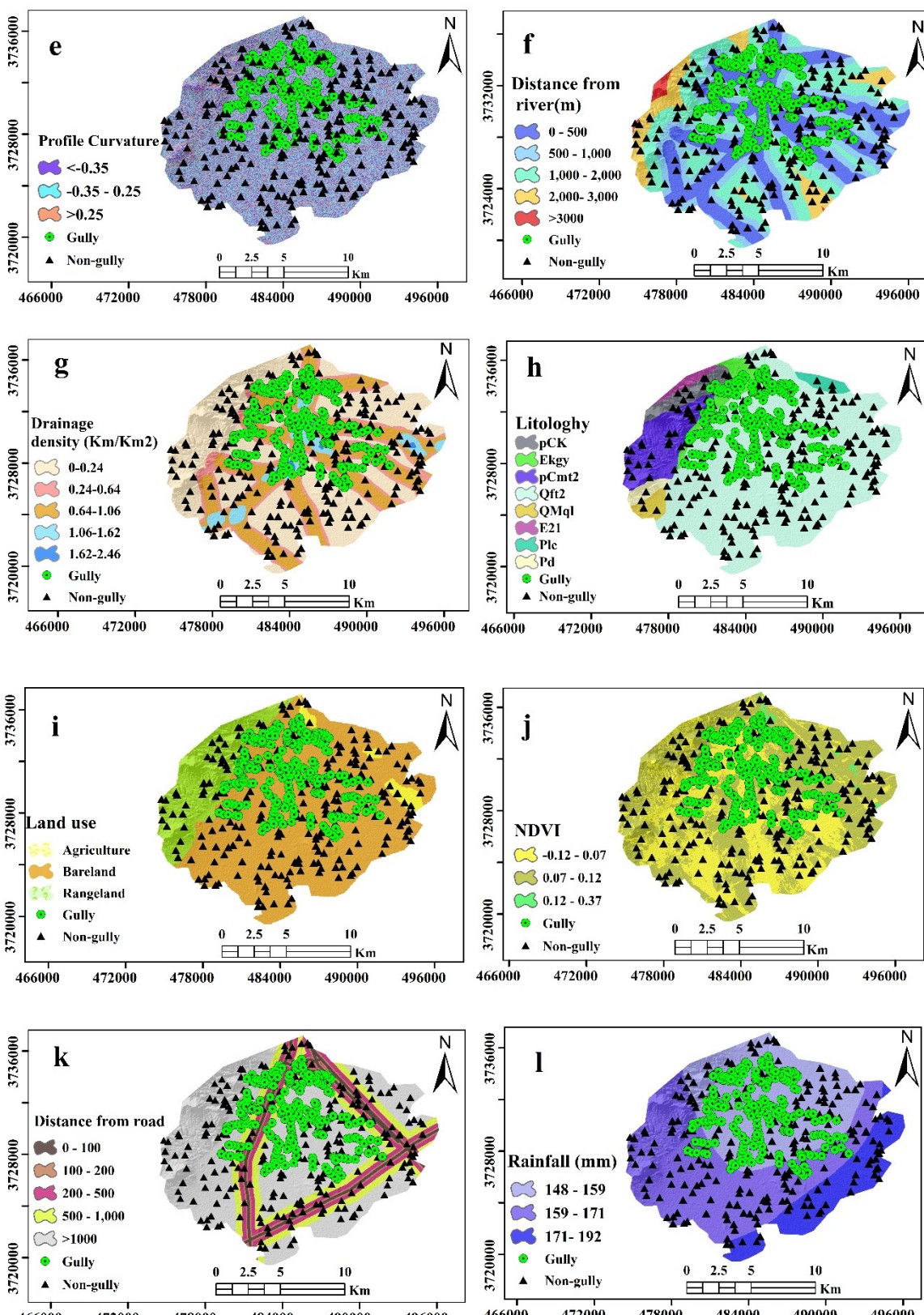

**Figure 4.** Gully erosion geo-environment factor maps of the study area: (**a**) DEM, (**b**) slope degree, (**c**) slope aspect, (**d**) plan curvature, (**e**) profile curvature, (**f**) distance to rivers, (**g**) drainage density, (**h**) lithology, (**i**) land use, (**j**) NDVI, (**k**) distance to roads, and (**l**) annual mean rainfall.

### 2.2.3. Gully Erosion Susceptibility Mapping Using Data Mining Methods

Random Forest (RF)

The RF technique is a modern, tree-based method that includes a multitude of classification and regression trees [56]. It is also a nonparametric method for modeling the continuous and discrete data of decision tree methods. The main problem faced by this method is the fluctuations in the results of each tree [41]. To reduce these fluctuations, and to reduce the estimation of variance, a random forest approach is proposed [57].

This is a combination of several decision trees that incorporate multiple bootstrap samples from the data, and a number of input variables randomly participate in the construction of each tree [58]. By using the bootstrap method, a large number of n samples from the initial observational data set are inserted [59]. During the sampling process, about one-third of the data as out-of- bag (OOB) was used for validation of models. Then, a tree is expanded based on any bootstrap sample [32,60]. During the process of constructing a tree in each branch, from between all $M$ independent variables, $m$ variables were randomly chosen for partition. For regression, the ratio $m/M$ is one-third, and is proposed for classification as $m = \sqrt{M}$ [61]. After introducing the whole tree construction, a number of trees are used as inputs and to determine the output [61]. By averaging these outcomes, the final output of the model is calculated, considering the empirical distribution of outputs, the percentile values, and the range of uncertainty.

In this research, a random forest model was computed in the R 3.3.1 software (developed by R core team located in University of Auckland, Auckland, New Zealand) using the "Random forest" package [62]. Then, the ArcGIS software was used to compute the gully erosion susceptibility, while the Gini index was used to calculate important factors in the R 3.3.1 software [63]. As a classifier, the RF makes an implicit feature selection, using a small subset of "robust variables" for classification only, resulting in superior performance in the subsequent data. The result of this selection of the implicit attribute of the random forest can be visualized with "Ginni index", and can be used as a general indicator of the significance of the attribute.

K-Nearest Neighbor (KNN)

K-Nearest Neighbor is in the class of algorithms that can classify an unknown entity if we have data with specific properties (X) and the value of the relationship (Y) [64]. The KNN Classifier is a sample and nonparametric learning algorithm. In the classification setting, the algorithm calculates the distance of the target point to the closest points according to the value specified for K, and according to the maximum number of votes of these neighboring points, in relation to the number of points was chosen [65,66]. By bypassing the density subordinate and going directly to a decision rule, the KNN algorithm supposes that pixels near each other in the trait space ought to fall into one class [67].

This method is based on the calculation of the similarity (neighborhood) of the real-time prediction value $X_r = \{x_{1n}, x_{2n}, x_{3n}, \ldots, x_{mr}\}$ to the predictive value for each historical observation $X_t = \{x_{1b}, x_{2b}, x_{3b}, \ldots, x_{mr}\}$ through the Euclidean distance function $(D_{rt})$ as follows:

$$D_{rt} = \sqrt{\sum_{i=1}^{m} w_i (x_{ir} - x_{it})^2}, t = 1, 2, \ldots, n \tag{2}$$

where $w_i (i = 1, 2, \ldots, m)$ is the weight of predictors, whose sum is equal to one.

### 2.2.4. Assessment of Data Mining Based Models

All gully and non-gully points in this area were classified into two categories to create gully erosion susceptibility maps: one group for modeling and one for validation. The accuracy and performance of the maps generated by the RF and KNN models were confirmed using a receiver operating characteristic (ROC) curve. The ROC curve represents the ability of a model to forecast the

occurrence and non-occurrence of a gully. The threshold-independent performance measure is the area under the receiver operating characteristics curve (AUC), which is used in various studies to assess the predictive value of the model [68,69]. To calculate the ROC curve, missing points (for example, places without gullies) were determined using the random extraction algorithm in ArcGIS 10.5, and these points were controlled in the study area not located in gully erosion. The range under the ROC curve was computed for the rate of success and the prediction rate for the gully sensitive maps [70]. The AUC values ranged from 0.5 to 1, whereby the higher the AUC, the higher the model's veracity [43].

## 3. Results

The KNN model was fitted and modeled in the R 3.3.1 software using the "CARET" package [71]. In this algorithm, there is a basic parameter, K, or the number of neighbors that needs to be optimized; this parameter specifies the voting system in the KNN algorithm in which the neighbors K are selected with the least distance, and then specifies the output model. For this reason, the number k, between 5 and 45, is optimized based on the accuracy of the model [65,66]. A final k value of 13 was obtained with an accuracy of 0.736 (Figure 5). Also, when assessing the effect of input variables on modeling, the results showed that rainfall, altitude, distance to rivers, and drainage density are the most important factors in the modeling process. In contrast, profile curvature, slope aspect, and plan curvature were recognized as the minimum significant factors in the modeling process, whereby plan curvature has no significance in modeling gully erosion susceptibility (Table 2).

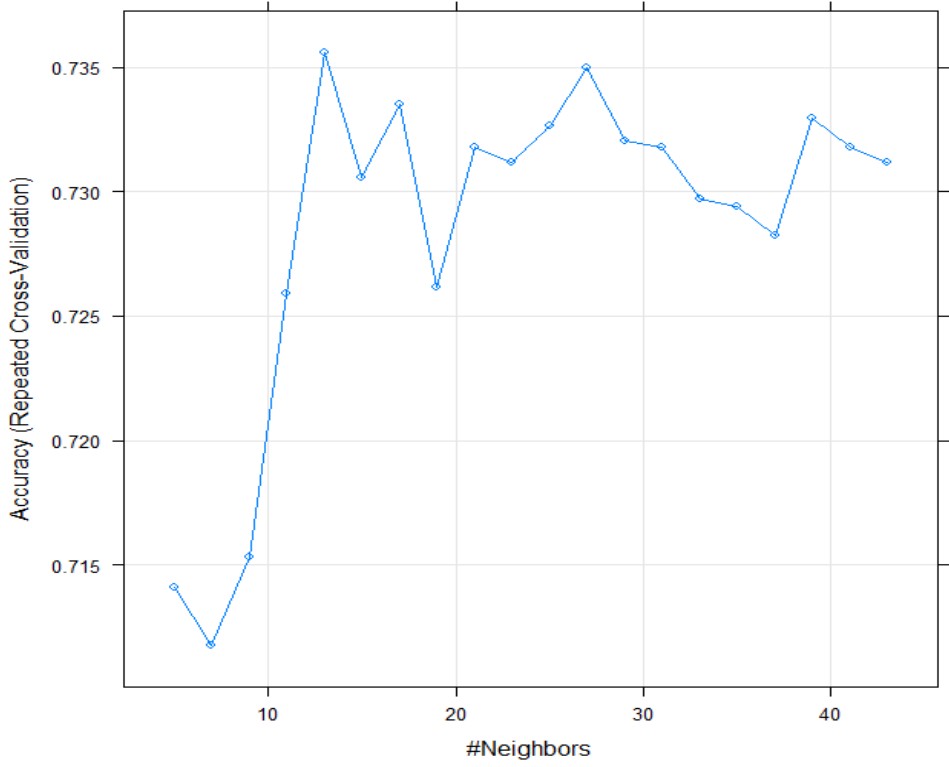

**Figure 5.** The results of cross-validation in the KNN model.

A confusion matrix shows the performance of the random forest model in the training stage; this matrix is not used in model evaluation, but rather, when the accuracy of a group detection is more important than the overall detection accuracy. The results of the confusion matrix for the RF model are depicted in Table 3. Based on Table 3, the model predictions are shown with "1" and the observations with "0". The results show that the training dataset and the model agree that there are no gully erosions for 137 observations, while there will be gully erosions for 145 observations. Nevertheless, there are 25

gully erosion pixels that the model predicts that are not gully erosions. Similarly, the model predicts that 33 observations will be gully erosions, where, in fact, they are not gully erosions.

The results of the RF prioritization using Gini index are depicted in Table 2. The results show that rainfall (48.74), altitude (30.46), distance to roads (18.36), and distance to rivers (14.95) have the highest importance scores.

**Table 2.** Share of the gully erosion efficacy variables to the KNN and RF methods.

| Variable | Importance | |
|---|---|---|
| | KNN | RF |
| Rainfall | 100.00 | 48.74 |
| Altitude | 74.35 | 30.46 |
| Distance from rivers | 50.64 | 14.95 |
| Drainage density | 30.11 | 6.40 |
| Distance from road | 19.39 | 18.36 |
| Land use | 17.56 | 2.18 |
| NDVI | 5.66 | 8.92 |
| Slope | 5.63 | 6.32 |
| Lithology | 4.54 | 4.07 |
| Profile curvature | 1.23 | 2.70 |
| Slope aspect | 0.92 | 4.82 |
| Plan curvature | 0.00 | 4.94 |

**Table 3.** Confusion matrix of the random forest (RF) model (0 = no gully, 1 = gully).

| Observation | Predicted | | Class Error |
|---|---|---|---|
| | 0 | 1 | |
| **0** | 137 | 33 | 0.1941 |
| 1 | 25 | 145 | 0.1470 |

Table 4 shows the results of determining the best parameter in the random forest model in the training phase. The results of *tuneRF* indicated an *mtry* of 5 whereas the highest accuracy find at ntree of 235.

**Table 4.** Extracting best parameters in Rf model.

| Node Size | mtry | Trees | Best Tree |
|---|---|---|---|
| 5 | 5 | 700 | 235 |

Figure 6 indicates the misclassification/error rate as a function of trees grown in the random forest. The black line represents the entire sample (out-of-bag) and the green line represents the error rate, where *non-gully* = 0 and the red line represents the error rate when *gully* = 1.

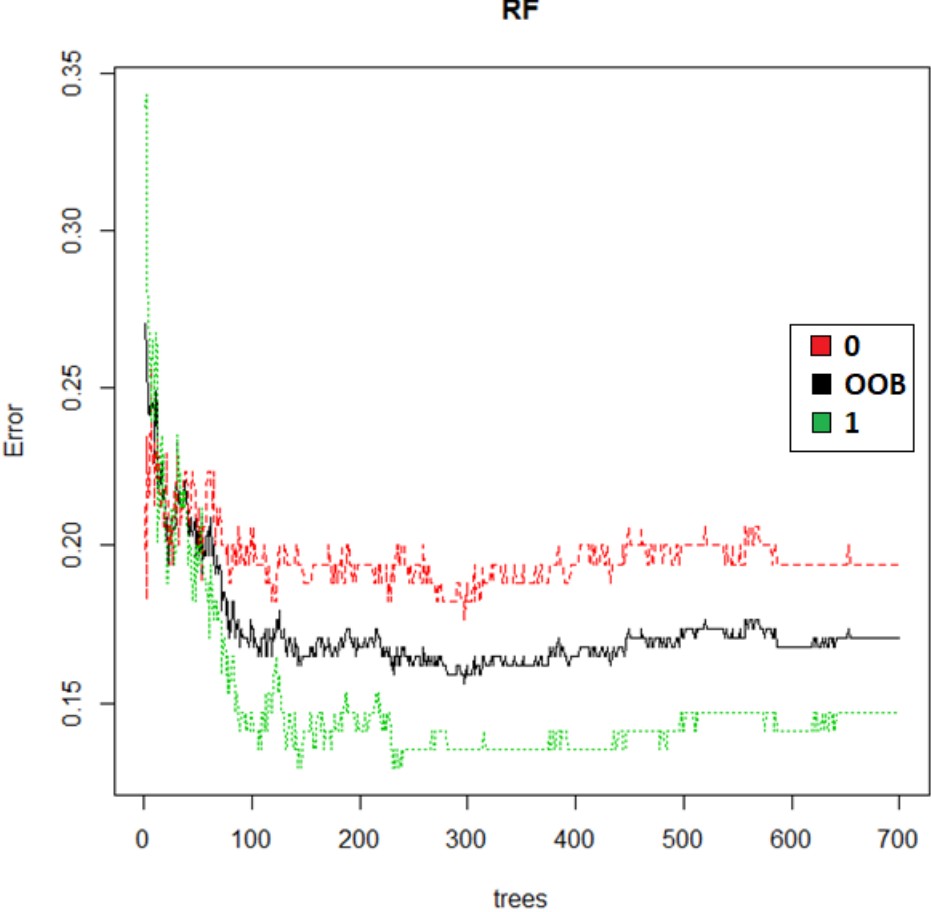

**Figure 6.** Misclassification/error rate as a function of trees grown in RF.

Finally, the gully erosion susceptibility maps were classified into four classes, namely, low, moderate, high, and very high susceptibility, using the "natural breaks method" in ArcGIS 10.5 (Figures 7 and 8; Table 5) [27,28]. The natural breaks classification system is a method for classifying data that is designed to optimize the order of a set of values to natural classes. A natural class is the most desirable class range naturally found in a dataset. Class range consists of items with similar properties that form a natural group in a dataset. This classification method seeks to minimize the mean deviation from the mean of the class while maximizing the separation from other groups. This method reduces the variance within classes and maximizes the variance between them. [72].

**Table 5.** Percentage distribution of susceptibility classes in the KNN and RF models.

| GPM Zones | RF | | KNN | |
|---|---|---|---|---|
| | Range | Area% | Range | Area% |
| Low | <0.217 | 46.42 | <0.2 | 13.23 |
| Moderate | 0.217–0.45 | 15.42 | 0.2–0.5 | 10.83 |
| High | 0.45–0.677 | 19.99 | 0.5–0.8 | 42.16 |
| Very high | >0.677 | 18.18 | >0.8 | 33.78 |

Based on the results of the KNN model, the classification of the susceptibility map resulted in the following class shares: low (13.23%), moderate (10.83%), high (42.16%), and very high (33.78%). On the other hand, the RF model resulted in different class coverage shares for the study area, namely, low (46.42%), moderate (15.42%), high (19.99%), and very high (18.18%).

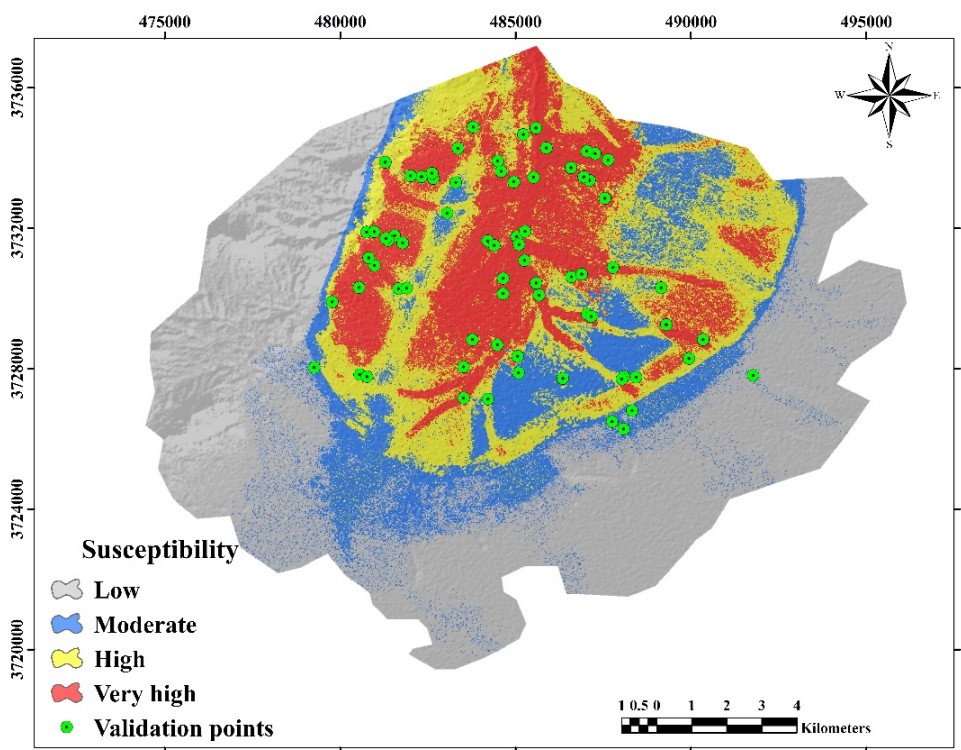

**Figure 7.** Gully erosion susceptibility map using the RF model.

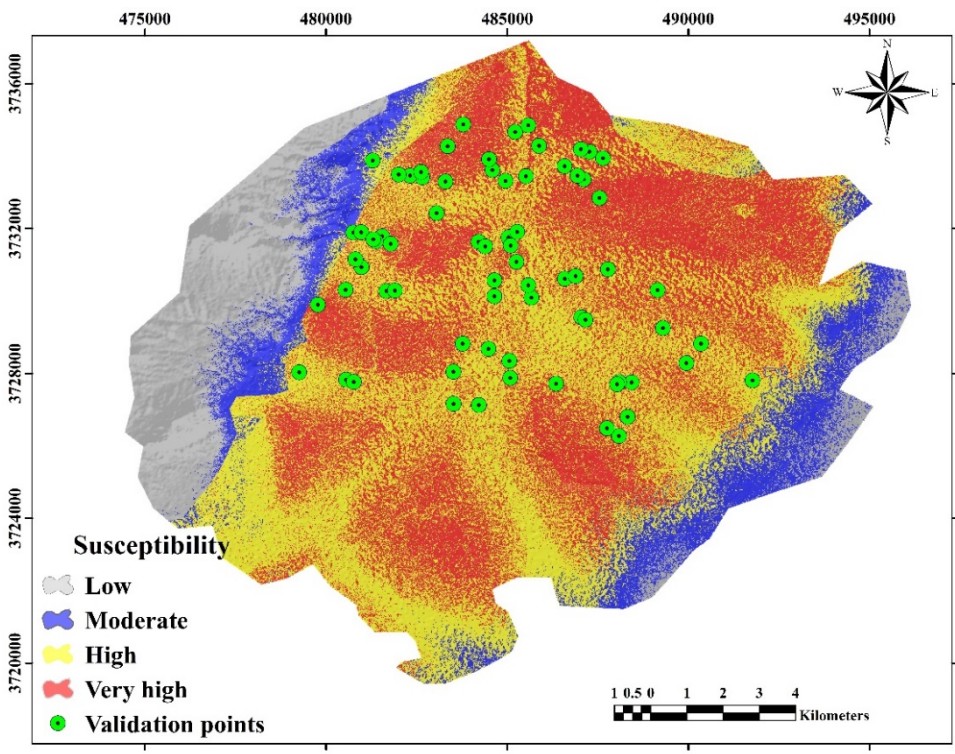

**Figure 8.** Gully erosion susceptibility map using the KNN model.

*Validation of Gully Erosion Susceptibility Maps*

The results of the AUC/ROC validation indicated that both models produced good results. For the KNN model, the AUC and prediction accuracy were computed to be 80.9% and 0.809, respectively;

in contrast, for the RF model, the AUC and predictive accuracy values were 87.4% and 0.874, respectively (Figure 9).

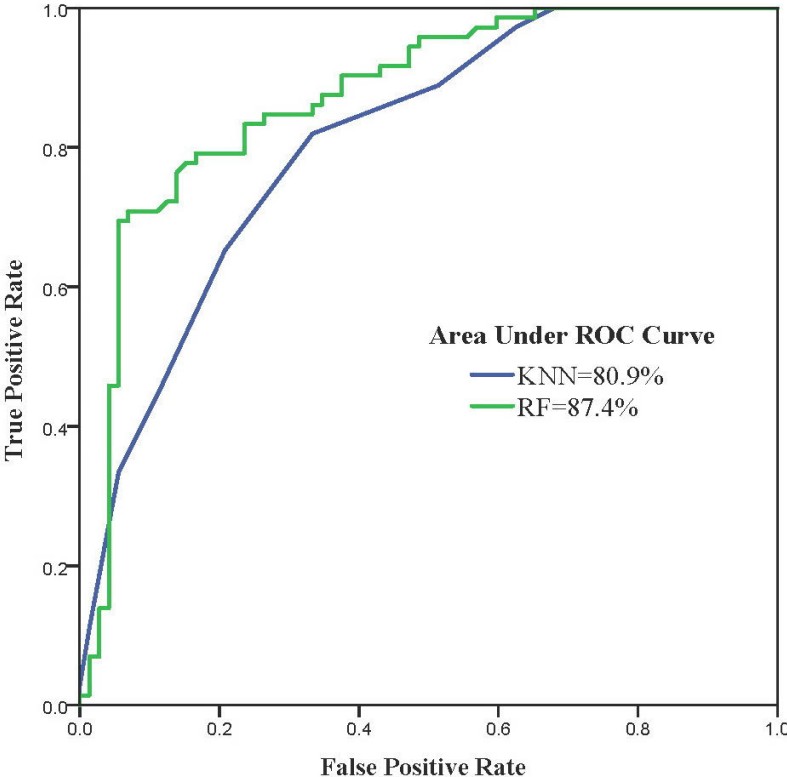

**Figure 9.** The ROC curve of the KNN and RF models used to map the study area.

## 4. Discussion

In general, our results show that the RF and KNN data mining models have a reasonable accuracy for gully erosion susceptibility mapping in our study area, whereby the RF outperformed the KNN based on the AUC values. These results are consistent with the results of previous studies [42,73–76] which suggested that the RF model is a robust and well-functioning model. RF is an advanced technique in spatial sciences. One of the benefits of the RF model is that it can support multiple input variables without deleting every variable and several other sets of classes that have high prophetic accuracy [41,77]. The classification accuracy of this model is influenced by several factors, such as range, scale, file type, and the accuracy of the computer file. The RF model has the ability to use explanatory variables in the modeling process [77]. The accuracy of RF modeling depends on various factors such as quantity, scale, type, and the accuracy of the input data [41]. This technique can manage and model a large data set [77] and can combine duplicate predictions of any phenomenon using multiple tree algorithms. The RF model can identify and notice nonlinear relationships between independent and dependent variables [41,78]. Therefore, utilizing all the acceptable factors for a modeling task will increase the accuracy of the model. Compared to alternative models, the RF has a strong ability to validate an outsized range of data sets [41,77]. The RF model has the power to assess environmental problems and hazards for any given area. The advantages of the RF model include the fact that it has robust and accurate machine learning algorithms, is considered a highly accurate classifier for many datasets, it can run efficiently on large databases, manage enormous of input variables without elimination, estimate effective factors in the classification, generate an internal, unbiased estimate of the generalization error as the forest building progresses, is an effective method for estimating missing data, and retains validity when a big portion of data are missing [79].

This study compared the results of the RF and KNN models for gully erosion susceptibility modeling, and determined that the KNN model has a lower validity than the RF method. The accuracy of the KNN model can, hence, be severely degraded by the presence of noisy or irrelevant features. In other words, the model structure is determined by the data. Naghibi et al. [80] and Naghibi and Moradi Dashtpagerdi [60] also showed that the KNN model has a lower efficiency than other methods in groundwater potential modeling.

The outcome of the variable importance validation indicates that rainfall is an effective variable for modeling gully susceptibility. This seems possible because rainfall is the basic source of runoff, and an initiating factor in gully erosion. Due to the high amount of rainfall in the region in December and the low vegetation cover, a rapid increase in gully erosion occurred in the area [3,76]. Rainfall effects on land can cause runoff and soil erosion. [81,82]. Also, the spatiotemporal heterogeneity of rainfall plays an effective role in erosion [83]. Because the spatial distribution of rainfall is irregular in the studied region, the rainfall regime is such that rainfall intensity is high, and its duration is short. These factors have led to an increase in erosion and contributed to the deployment of the gully. Also, it was observed that altitude is another important factor in this study. Based on the results obtained, the locations at the lowest elevation are most susceptible to gully erosion, which is consistent with the finding of [42]. This may be due to a gully erosion mechanism (incision, seepage, or piping) that is more likely to occur in lowland areas. Therefore, this factor could be considered as a distinguishing variable in the susceptibility mapping of gully erosion. Areas closer to rivers have more developed drainage systems, which increases the water flow and speed, thereby resulting in gully erosion. This fact is reflected in the results of this study, as it is the third most important factor in the modeling procedure. And, by visually inspecting the gully susceptibility map obtained by the RF model, it can be seen that areas closer to the river system have a higher susceptibility to this erosion type. Also, this can be explained by the fact that the river, with its underlying action and erosion, disturbs the balance of the slopes overlooking the waterway, increasing the sensitivity to gully erosion along the river's edge. These results are consistent with the results of [22,84]. Dube et al. [85] also obtained similar results regarding the inverse influence of distance from a river on gully susceptibility. Roads are anthropogenic construction projects that artificially concentrate surface runoff and increase its speed could be an intensifying factor in gully erosion. This fact has also been observed and supported by the results of the present study. Roads concentrate surface runoff and runoff depression from other basins into the basin. Hence, gully erosion increases after the construction of a road [54].

## 5. Conclusions

Due to the destructive nature of gully erosion, researchers and natural resource managers around the world have concentrated on mapping the susceptibility and assessing the hazards of gully erosion. In this research, we employed the RF and KNN data mining models to assess the results of geo-environmental variables on gully erosion and to identify areas prone to this hazard. For this purpose, we evaluated twelve variables. The methodological framework used in this study has demonstrated that the appropriate choice of effective factors in gully erosion, along with the use of data driven techniques, can correctly identify gully erosion-prone areas. The expansion and advancement of gully erosion have severe environmental impacts, including the destruction of fertile surface soil, damages to roads, damages to canal routes, and the exhaustion of large volumes of rainfall as floodwater flows through the gully. In terms of human impact, the destruction of agricultural land caused by gully erosion can lead to increasing unemployment and migrating villagers, and, in some cases, entire villages need to be evacuated due to other damages caused by the expansion of the gully. Identifying and predicting which areas may be susceptible to gully erosion can help reduce the destructive effects and prevent the future development of this type of erosion, and can provide significant assistance to the people of the study area. The susceptibility maps prepared using the RF and KNN methods are suitable tools for appropriate projections to preserve lands from gully

erosion. Accordingly, this methodology can be used to determine gully erosion in other, similar regions, especially those with the same dry and semi-arid climate.

**Author Contributions:** Conceptualization, M.A., S.J., S.A.N. and H.R.P.; methodology, M.A., S.J., S.A.N. and H.R.P.; software, M.A., S.J., S.K.B. and H.R.P.; validation, M.A., S.J., S.A.N., S.K.B. and H.R.P.; formal analysis, M.A., S.J., S.A.N. and T.B.; investigation, M.A., S.J., S.A.N., S.K.B., T.B. and H.R.P.; writing—original draft preparation, M.A., S.J., S.K.B., S.A.N., T.B. and H.R.P., writing—review and editing, S.A.N., H.R.P., T.B.; project administration, M.A., H.R.P. and T.B.; funding acquisition, T.B.

**Funding:** The study was funded by the Austrian Science Fund FWF through the GIScience Doctoral College (DK W 1237-N23) at the University of Salzburg. Open Access Funding by the Austrian Science Fund (FWF).

**Acknowledgments:** The study was supported by the College of Agriculture, Shiraz University (Grant No. 96GRD1M271143).

**Conflicts of Interest:** The authors declare no conflicts of interest. The funders had no role in the design of the study, in the collection, analyses, or interpretation of data, in the writing of the manuscript, and in the decision to publish the results.

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
