# Peer review of "A Comparative Assessment of Random Forest and k-Nearest Neighbor Classifiers for Gully Erosion Susceptibility Mapping"

_water, doi:10.3390/w11102076_

Round 1

Reviewer 1 Report

Change the Thomas Blaschke reference in Authors

There is an updated review of the statistical methods of gully erosion assessment that introduces very well to the central theme of the manuscript They work with mining techniques such as RF and KNN both of wich heve not yet applied in this field and that is an important value in the results obtained , although it would be necessary to verify the certainty of such results in the field and specify to what extent they represent a good management measure in the treatment of land uses in this type of areas.

The gully susceptibilty map is an appropiate tool to understand the mechanism of gully erosion and the aid scientific planning and decision making. But the results of the areas should be checked in the field.
It would have been interesting to know some of the field measurements made in the studied area, such as showing in a photo the most outstanding geometric measurements

It would have been interesting to know the works that can be proposed in the near future after this work as the possibility of working with field plots where to establish paired catchments and obtain the origin of the gullies

DEM with better definition could be obtained in cells smaller than 12.5m, working with other images that provide other platforms such as PNOA
In Figure 5 with 20 points in the graphic we could comment something more than those expressed in the explanation of the paragraph (line 214) where it is not clear: “A final k value of 13 was obtained with a veracity of 0.73…”

The conclusions show some socio-economic aspects that have nothing to do directly with this purely research work applied to two tools and whose results must be applied by river basin managers. They are obvious aspects in Iran as in any other part of the world and should not be a conclusion of it. All this is indicated at the beginning in the introduction that any specialist knows

Author Response

Thank you so much for your positive comments.

Best

Reviewer 2 Report

The article is a good base for creating the really good one, but I suggest to consider following major issues:

The description given in 2.2.2 has to be corrected to the more understandable form. It seems that many types of input data were obtained from the dedicated software. What was the input to the software, how it was obtained (where from? by whom?), which features have been obtained via the output of the software? It can be described more precisely.

Part of the features is described with units, but the rest not. For some of them, only the lowest and the highest are given without borders between classes. Does the distance to the river mean the closest one? Drainage density is given in km per square km - it needs some more explanation.

There is one big misleading issue: in lines 201-202 it is described that gully points were divided and used for modeling and validation. What about non-gully points? Were they used for training too? And for validating? The text says "no", but ROC curves and confusion matrix suggest that they included in training and validating sets. It has to be clarified and clearly described. Moreover, in lines 207-209 it is written that non-gully points were chosen by the algorithm in the software. Are the authors sure that there are no gully points? Can the reader be sure too? It is crucial for the meaning of the whole article. The number of samples for training, testing and validating purposes have to be clearly stated (see specific remarks below).

It has to be clearly stated for both methods what is the output. It can be read in the following part (but not in chapter 2) that the output from RF is classification gully/non-gully. It has to be clearly described in chapter 2. The same is for KNN, but there is no confusion matrix for this tool. Why?

It is stated (line 186), that for RF Gini index is applied to calculate importance factors. They cannot be found in the article. The prioritization (written in line 226) mechanism is unknown. The discussion of 100 efficacy (for rainfall in KNN) should be discussed. What does it mean? Is rainfall in this method sufficient for predicting gully point?

Again, how to come from classification gully/non-gully to susceptibility? The answer in the article is: dedicated software did it. Even an idea of calculating susceptibility should be presented.

Discussion (chapter 4) is more about applied tools and their potency. I suggest discussing the results achieved. The decision of applying RF and KNN should be proved in the Methodology chapter. Discussion is for discussing results achieved.   

Why the authors say about "flood susceptibility" (line 273) while they investigated gully erosion susceptibility.

In lines 69-70 the authors stated that one of the purposes of the article is to "provide applicable guideline for stakeholders to reduce gully damages in the study area". I can find such a guideline.

Some specific remarks:

Figure 1: if the contour map is taken from any web site, please show the source as reference; usually, red colour is used for higher part of land, green for lower - when applied oppositely, it makes the reader wondering what is the real shape of the terrain; what is "DEM map"? (in the caption).

Figure 2: are the photos made by authors? if not, please reference them

Figure 3: while fig. 3 is mentioned in chapter 2.2 it should be located there (in the present form it is one-sentence introduction)

Line 98: it is better to apply the third form (not "we") or any form (e.g. research was conducted); I would be nice to know if authors have found and monitored these gullies personally (if not the reference should be added); when these investigations were done?

Lines 100-102: the sentence about the origin of gullies has to be supported by a reference

Line 105: "The same number" i.e. 72? 170? 242? Please clarify

Line 115: "DEM map" ???

Line 130: the web site address should be presented as reference

Lines 140-141: the sentence: sth depends on features of surface material and the surface of Earth - has to be corrected (shape of the surface?)

Line 141: What is the source of the geological map? Please reference it.

Line 147: What is the source of the input to the software? Please reference it

Lines 149, 151: "NDVI map" ? please explain

Line 153 and formula (1): IR and R - are they: length of the waves or frequencies? How to subtract the band from the band?

Lines 154-155: I understand that the existence o a road can influence the occurrence of gully erosion, but canals and roads can't be susceptible to the erosion; the sentence should be reformulated

Line 160: IDW - please explain

Lines 208-209: The range under ROC curve (...) for each gully sensitive map. The sentence has to be reformulated. In a present form is not understandable.

Lines 213-214: the reference (Kuhn, 2008) should be replaced with the correct reference format

Lines 214-215: stated coefficients are not presented in stated figure 5. Their meaning is not explained and unknown

Lines 226-227 and Table 2: 30,45 or 30,46 (compare text and table)? The influence of Rainfall is omitted in the text for RF

Line 280: please correct "plausible"

Author Response

(The authors gave the same response as above.)

Reviewer 3 Report

Referee report on manuscript «A comparative Assessment of Random Forest and k-Nearest Neighbor Classifiers for Gully Erosion Susceptibility Mapping» by Mohammadtaghi Avand et al.

The manuscript compares and evaluates two different approaches for elaborating gully erosion susceptibility maps. The comparison is interesting and the results provide valuable results for modelers. The manuscript is in principle well organized but has a few flaws that should be addressed before publication:

General remarks:

In my opinion, the authors do an assessment of two methods and not a validation. Thus, I recommend changing line 22 in the abstract (“We assessed [..]” instead of “We validated”. The same is valid for chapter 2.2.4. The analysis of the ROC is not a strict validation but an assessment. Please change the title of this chapter.

Regarding the validation, the authors state in figure 3 that they use 30% of the dataset for validation. However, this is not described in the further chapters. I absolutely recommend a strict validation of the trained model (trained with the 70% subset) with the 30% subset. The lacking validation is my main criticism.

Please describe the final structure of the trained models (in a graph or in another form)

Line 281: The reasoning on the relationship between annual rainfall sum and rainfall maximum intensity is not sound. Please check or state this with references.

Line 288: your explanation on the altitude is vague. If a certain speed of runoff is needed for erosion than the slope must be a better predictor than altitude.

Lines 315-316: This statement is rather speculative and not resulting from your studies. Please reformulate or add a reference.

Minor remarks:

Thomas Blaschke is from institution no. 5 (affiliations)

Line 19: we determined

Line 65-68: This formulation is a bit unlucky: First, you state that SGT is a “good” theory for erosion prediction and then you state the purpose of your research: Testing RF and KNN. The first sentence is not guiding the reader to your purpose but it disrupts the logic of the introduction.

Line: 68 “famous” is not a criterium for choosing an approach to solve your research question

Line 74: is useable

Chapter 1.2 study area should be numbered 2.2

Line 149: for non-remote sense experts please describe very shortly NDVI (write out the long name of this index before using the abbreviation)

Line 196: class

Line 203: describe in short the principle of the ROC (not using only the abbreviation)

Results: table2: What is with your non-gully locations as stated in lines 105-106. Why did you not use these points for validation and assessment? The false alarms are very interesting for assessing models

Line 253: These results are consistent

Line 254: advanced

Line 258: Please describe how a file type and the accuracy of the computer file can influence your results. This cannot be the case. Maybe there is a misunderstanding, please reformulate.

Line 310: What is scientific prosperity?

Line 311: Please avoid subjective valuations as for example “impressive”

Author Response

(The authors gave the same response as above.)

Round 2

Reviewer 2 Report

Almost all remarks have been considered. But some (some of them very important) issues have to be clarified. 

Equation (1) for NDVI is still not explained. What is the unit of the band? Wave length? Frequency? It is not a subject of the paper i.e. intensity of green cover. It is better to omit sth than giving not precise information.

If "IDW is one of the most applicable..." (line 173 please describe it or reference it. Show the full name of IDW.

Line 238. k value appears the first time in the paper. Its value of 13 was obtained with the stated accuracy??? Some more description is necessary.

Comparing KNN and RF by ROC curves is understandable. But it is not understandable why the identical measure for classification accuracy can't be applied for both these tools. Prioritization of the strength of the independent variables (features) is a secondary issue for map creation.

The method which allows coming from automatic classification to probability map is still not explained. "natural brakes method" included in the software is stated. I - as a reader - would like to know even an idea of this method. It is not even referenced. It is crucial for the key calculations made in this paper.

There is subchapter 1.2 in chapter 2.

Author Response

Dear Reviewer

Thank you so much for your positive comments.

Please see attached file and also your replies in original file as highlighted.

Best

Round 3

Reviewer 2 Report

Line 165: NIR? It should be IR.

It is still unknown what are you adding in the formula (1). Lenghts of the bands? Frequencies? "imargery"?

Why does NDVI vary then?

"withen" - Do you mean "within"?

Author Response

#The author would like to thank the Associate Editor and reviewer(s) for their comments.

Comment: Line 165: NIR? It should be IR.

Reply: Modified and added to text.

Comment: It is still unknown what you are adding in the formula (1). Lenghts of the bands? Frequencies? "Imagery"?

Reply: Frequency: Frequency describes the number of waves that pass a fixed place in a given amount of time. Usually frequency is measured in the hertz unit.

Wavelength: The wavelength is the spatial period of a periodic wave the distance over which the wave's shape repeats. It is the distance between consecutive corresponding points of the same phase on the wave, such as two adjacent crests, troughs, or zero crossings, and is a characteristic of both traveling waves and standing waves, as well as other spatial wave patterns.

Brian Hilton Flowers (2000). "§21.2 Periodic functions". An introduction to numerical methods in C++ (2nd ed.). Cambridge University Press. p. 473. ISBN 0-19-850693-7.

Raymond A. Serway; John W. Jewett (2006). Principles of physics (4th ed.). Cengage Learning. pp. 404, 440. ISBN 0-534-49143-X.

Imagery: https://www.esri.com/about/newsroom/arcnews/landsat-8-imagery-available-for-online-users/

Comment: Why does NDVI vary then?

Reply: In this equation pixels values in these two different bands are used, not their wavelengths or frequencies.

In order to have a standard index with known minimum and maximum values for vegetation cover, NDVI is proposed. And according to the nature of the equation, it results may vary between -1 and +1. 

And the reason that NDVI varies is due to the fact that that pixels values in each band are different.

Comment: "withen" - Do you mean "within"?

Reply: Modified in text

Thank you so much for your positive viewpoints in this article.